# OpenReview forum: "Non-Euclidean Gradient Descent Operates at the Edge of Stability"
_ICML.cc/2026/Conference — ICML 2026 spotlight_

### Official Review · Reviewer_1EwL · 2026-03-03

**Soundness:** 3
**Presentation:** 3
**Significance:** 3
**Originality:** 2
**Overall Recommendation:** 5
**Confidence:** 4

**Summary:**

Considering gradient descent as the solution of the regularized linearization around the current point $w_t$ in which the norm of the difference between $w_t$ and the new point is not necessarily the $\ell_2$-norm, non-Euclidean gradient descent methods are all those updates obtained by selecting different norms. The authors recall a few options and corresponding methods used in the literature: $\ell_\infty$ norm, block $\ell_{1,2}$ norm, preconditioned norm, and spectral norm on the matrix-shaped parameters. The paper shows how to extend the edge of stability observations to different non-Euclidean gradient methods. In particular, the authors derive the corresponding generalized sharpness and, being this quantity the solution of a quadratic convex-constrained optimization method, also the Frank-Wolfe iterations for each of the above-mentioned norms. The numerical experiments show that the values computed by Frank-Wolfe are better suited than the classical $\ell_2$-sharpness to capture the edge of stability phenomenon for non-Euclidean gradient methods. Beyond the classical updates, the paper also covers various normalized updates and their generalized sharpness.

**Compliance With Llm Reviewing Policy:**

Affirmed.

**Final Justification:**

The authors have addressed all my concerns, I am keeping my original score.

**Key Questions For Authors:**

4. Why are the predicted value of directional smoothness and generalized sharpness less precise in case of transformers architectures? Why are they often trained with larger step sizes?
5. How where the fixed step sizes chosen? Overall, the description of the experiments could be improved (e.g., dimension of MLP, CNN).
6. Why transformers architectures are only trained with 2 different step sizes and MLP/CNN with 3?


**Minor remarks**

7. row 104-106 (left column): not a real contribution, the authors should rephrase the main contribution of Section 5.

8. r 202 (l): "flattening all parameters of the networks" probably means "considering all parameters of the network as a single vector", right?

9. r 218 (l): just "Newton's method" is misleading, the Hessian needs to be positive definite, the authors could add something like "for convex functions".

10. Figure 11: what is the step size used in this experiment?

11. Figure 18: did the authors run their sharpness experiments on full-batch and full-size CIFAR10? This sounds enormously expensive, could the authors quantity the runtime of these experiments? Could the authors recall the motivation for doing so?

12. r 1199, 1362: fix "Figure ??"

**Limitations:**

To transition from the directional smootheness defined between two consecutive points to the sharpness in a single point, the authors assume that the Hessian is almost constant over the line segment originated in $w_t$. This assumption is generally not true for neural networks, sometimes not even for very small $\eta$, however, it also not really needed to define the generalized sharpness. I would suggest the authors to simply remove this sentence.

**Strengths And Weaknesses:**

The paper provides a complete framework for non-Euclidean gradient methods and their generalized sharpness. Moreover, it nicely combines both aspects of theory and practice. In fact, I find the idea of using Frank-Wolfe to actually plot the generalized sharpness to be a valuable contribution.

**Weaknesses**

1. The name suggested by the authors for the quantity $D^{||\cdot||}(w_k, w_{k+1})$ is misleading. This is not a quantity that measures only the directional smoothness, as it admits negative values (how does one interpret a negative directional smoothness?). This value is actually obtained by solving for $L$ the Descent Lemma, as previously done in [1]. For this reason, it is not surprising that if $\Delta L_t \leq 0$, then $D^{||\cdot||}(w_k, w_{k+1})\leq 2/\eta$. As a consequence of the chosen nomenclature, the claim of Contribution 2 is also misleading. If one does not read the rest of the paper, this sentence leaves with the impression that the authors were able to prove a Descent Lemma that holds in both directions.

2. The correlation between $D^{||\cdot||}(w_k, w_{k+1})$ and the sharpness (Contribution 1) was already identified in [1].

3. While the paper provides a very clean framework for non-Euclidean methods and their generalized sharpness, the conclusions that can be drawn from it do not exceed expetation.

[1] Fox, C., Galli, L., Schmidt, M., & Rauhut, H. (2024). Nonmonotone Line Searches Operate at the Edge of Stability. In OPT 2024: Optimization for Machine Learning.

---

> ### Author Rebuttal · Authors · 2026-03-31
>
> Thank you for recognizing our unified framework for non-Euclidean gradient methods and generalized sharpness, as well as the balance between theory and practice. We value the positive feedback on using Frank–Wolfe to estimate and visualize generalized sharpness. We refer to weaknesses, questions, remarks, and limitations by W, Q, R, L
>
> ### Weaknesses
>
> W1: Indeed, contribution 2 isn't clear enough. We will rephrase it as follows: "We show that, by extending the definition of the directional smoothness to allow for negative values, we establish that the loss will decrease if and *only if* $D^{\|\cdot\|}(w_k,w_{k+1})\le 2/\eta$." As for interpreting negative values for directional smoothness, a negative value means there is a *strong* direction of descent. Geometrically, this means there is negative curvature (and negative eigenvalues of the Hessian), which leads to descent. This is natural for DL tasks, which are non-convex and can have negative curvature directions. As for the name "directional smoothness", perhaps "directional curvature" would be better to accommodate for possible negative values? Since negative curvature is an accepted notion, whereas negative smoothness is not
>
> W2: Thank you for providing this reference. Our reading of [1] is that it studies line search techniques in the context of Euclidean norm, and didn't found no definition that could be interpreted as directional smoothness. Could the reviewer please clarify where the notion of directional sharpness is used in [1]?
>
> W3: Could you please elaborate? Without an objective remark, we are not sure how to respond. Regarding our own expectations as to the conclusions, we hoped to be able to identify the correct notion of sharpness for non-Euclidean geometries. Our experiments indicate that we have
>
> ### Questions
>
> Q4: We connect these issues to CE loss used for transformers. Problems with measuring EoS for CE loss were also observed in Cohen et al, 2021 (see Sec 3.1 for more details). They tested the training of a model with MSE and CE losses and found that training with CE loss does not reach EoS "because the second derivative of the cross-entropy loss function (which appears in the Gauss-Newton term of the Hessian) is small when the margins of the classifier’s predictions are large". In our case, we had to choose a large enough LR to both display EoS and still be convergent. This may, in part, explain why the generalized sharpness and directional smoothness are so noisy
>
> Q5: We will add a more detailed description of the MLP and CNN experiments: MLP architecture contains 3 hidden layers of size 64, and CNN architecture has 2 convolution layers of size 128. $\eta$ is chosen so that training converges and is large enough to have an EoS phase. In particular, we measure the dual gradient norm for a few iterations in the beginning to understand what range of $\eta$ will work for longer runs
>
> Q6: To address this concern, we added additional LR so that all setups have 3 LR; see **[[$\ell_{\infty}$](https://anonymous.4open.science/r/EoS_Figures-C8F6/L_inf_descent.png)]** and **[[Spectral](https://anonymous.4open.science/r/EoS_Figures-C8F6/SpectralGD.png)]**
>
> ### Remarks
>
> R7: Indeed, our statement that we "analyse the quadratic case" is not a clear contribution. Instead, we will rephrase the main contribution of Section 5 as the following: "We validate our insights of generalized sharpness $S$ on quadratics, and prove that for $\eta < 2/S$ the corresponding non-Euclidean gradient descent converges, and that there exists initiliazations for which $\eta >2/S$ lead to divergence. This second point is fundamental for understanding the presence of oscillations in the loss function”
>
> R8: Indeed, by flattening, we mean to stack all columns of the matrices and vector parameters into one vector
>
> R9: We will clarify this by stating "This includes ... Newton's method for convex functions"
>
> R10: First, thank you for digging deep into the appendix! The stepsize is $\eta=0.5\cdot10^{-7}$, we will add this to the caption, and apologise for the omission
>
> R11: The experiments in Fig 18 are conducted on full CIFAR10 dataset, which requires substantial computational resources. Each run takes around 1.5–2 days on a single RTX 4090. We performed these experiments for two main reasons. First, they demonstrate results on realistically sized datasets, showing that Non-Euclidean EoS also emerges in practical settings. Second, when using only a subset of CIFAR10, both ResNet and VGG models tend to memorize the data, and Spectral GD minimizes the loss extremely quickly. In such cases, the loss drops to values close to $0$ within a few iterations, making it difficult to observe the EoS over an extended period
>
> R12: We thank the reviewer for careful reading. In line 1199, we refer to Figure 12, while in line 1362 we refer to Figure 18
>
> ### Limitations
>
> L: Indeed, Hessian can change drastically even between two close points. We will remove the sentence to avoid such confusion

---

> > ### Author Rebuttal · Reviewer_1EwL · 2026-04-01
> >
> > I would like to thank the authors for their detailed and honest response.
> >
> > W1: indeed, "directional curvature" seems more appropriate.
> >
> > W2: in [1], this quantity is called $L_{approx}$ and it is defined in Appendix A.2 only for classical GD, but independently from line search methods. The identified correlation is between the (again classical) sharpness and $L_{approx}$ (see Figure 3, top row).

---

> > > ### Author Response · Authors · 2026-04-06
> > >
> > > We thank the reviewer for the follow-up comments, that led to the improvement of our work. We will incorporate the necessary changes, including a discussion around [1], to the revised version of the paper.

---

### Official Review · Reviewer_EW6F · 2026-03-12

**Soundness:** 3
**Presentation:** 3
**Significance:** 2
**Originality:** 2
**Overall Recommendation:** 5
**Confidence:** 4

**Summary:**

The paper proposes a generalized definition of sharpness under non-Euclidean norms, extending the original sharpness measure used in prior work. To compute this quantity in practice, the authors employ an approximation algorithm based on the Frank–Wolfe method. Through experiments on several models and datasets, the paper shows that the newly defined sharpness metric also exists an edge-of-stability–like behavior during training.

**Compliance With Llm Reviewing Policy:**

Affirmed.

**Final Justification:**

Through the rebuttal process, the authors clearify the my questions toward different part of the approximation algorithm and the theoretical positioning of the work. Therefore, I would like to raise my score to 5.

**Key Questions For Authors:**

Questions:

1. Interpretation of non-Euclidean sharpness: How should sharpness defined under different norms be interpreted geometrically in terms of the loss landscape? For example, does a sharpness defined under an l1 or other non-Euclidean norm correspond to a particular directional curvature or anisotropic property of the loss landscape?

2. Accuracy of the Frank–Wolfe approximation: Is there a way to quantify the discrepancy between the FW-based approximation and the exact solution of the optimization problem defining generalized sharpness? For example, can the authors empirically compare the FW estimate with exact solutions on small models where the optimization problem can be solved more precisely?

**Limitations:**

yes

**Strengths And Weaknesses:**

Strength

1. Novel sharpness definition: The paper introduces a generalized sharpness metric defined under non-Euclidean norms which also exist edge of stability phenomenon.

2. Empirical evaluation across multiple architectures: The experiments include several model families (ResNet, CNN, and MLP, transformer) on the CIFAR-10-5k and language dataset, demonstrating that the proposed sharpness measure can be computed and tracked across different experiments.

Weakness

1. Approximate computation of generalized sharpness: The generalized sharpness is computed using a Frank–Wolfe (FW) approximation with a manually chosen stopping criterion. It is unclear how accurate this approximation is relative to more direct Hessian-based calculations used in prior work on sharpness and edge-of-stability analysis. The paper would benefit from validating the approximation quality, for example by comparing FW estimates with exact calculation on smaller models.

2. Limited novelty in the theoretical analysis: The theoretical results show convergence and divergence behavior under a quadratic approximation of the loss landscape. Such analyses are well studied in the optimization literature. Extending these results to different norm geometries is interesting, but provides limited new insight into the optimization dynamics beyond what is already understood.

3. Potential coordinate-change: It is unclear whether the observed phenomenon reflects fundamentally new optimization behavior or simply change of geometry induced by the norm. Under this view, the stability condition may follow directly from classical results with a modified operator. It would strengthen the paper if the authors clarified whether the proposed generalized sharpness leads to behaviors that cannot be explained by such coordinate transformations.

---

> ### Author Rebuttal · Authors · 2026-03-30
>
> We thank the reviewer for highlighting the novelty of generalized sharpness, its connection to EoS, and empirical validation.We use W and Q to refer to Weaknesses and Questions
>
> ### Weaknesses
>
> W1: We acknowledge that FW approximation may be inaccurate in some cases and study this in Sec F.2,G.2,H.2. First issue is that for both infinity and spectral norms we don't have a closed form solution, and computing the associated generalized sharpness is a NP hard problem. Thus, it is infeasible to exactly check the accuracy of FW approximation. Knowing this issue, we performed the following checks:
>
> - Sensitivity of FW to the number of restarts (Fig 11, 13, 15): for a sufficiently large number of restarts, FW approximation stabilizes around one value
>
> - FW approximation is greater than the known lower bound given by Rayleigh coefficient.
>
> - For Spectral GD (Fig 16), increasing number of steps beyond 5 doesn't affect estimate, indicating sufficient accuracy of FW approximation
>
> - With enough restarts, FW approximation and true value of generalized sharpness are closely aligned for Block GD (Fig 13), confirming that FW approximation is accurate enough
>
> W2: We kindly disagree with this statement. Convergence of non-Euclidean gradient descent on quadratics is significantly more challenging than that in the Euclidean setting, and isn't well studied in the optimization literature. For example, in the Euclidean case $\eta < 2/L$ is a **necessary and sufficient** condition for convergence. Because of this understanding, we know in the Euclidean setting that the loss must initially grow when $\eta > 2/L$, and decrease when $\eta < 2/L$ because of the local quadratic expansion
>
> The same understanding in the non-Euclidean setting isn't known, and is needed to understand the EoS phenomena. We demonstrate in the non-Euclidean setting in Theorem 5.1 that $\eta < 2/S$ is sufficient condition for the loss to decrease. This result is known, see for instance Section 2.1.1 in [1], though the end convergence result isn't clearly stated. But what wasn't known previously is our divergence result in Theorem 5.2, where we provide initializations when it is also a necessary condition. However, we don't yet know what happens beyond such set of initializations. That is, the necessary and sufficient conditions for convergence (and divergence) still aren't known for non-Euclidean gradient descent. Understanding this divergence on quadratics is needed to understand EoS, since the oscillation at EoS occurs because the local quadratic temporarily diverges
>
> [1] Balles et al, The geometry of sign gradient descent, arXiv:2002.08056
>
> [2] Karimireddy et al, Error feedback fixes signsgd and other gradient compression schemes, ICML2019
>
> [1] Nesterov, Introductory Lectures on Convex Optimization: A Basic Course, 2004
>
> W3: Please let us know if this is a correct interpretation of the question: "Can these notations of generalized sharpness be captured by L2 sharpness but under a change of coordinates?".
>
> Answering above, all non-Euclidean methods can indeed be written as vanilla gradient descent with adaptive preconditioner (thus a change of coordinates). But preconditioner depends, in a highly non-linear way, on the current gradient. It doesn't correspond to applying vanilla gradient descent with a simple linear change of coordinates. Instead, the coordinate system changes at every iteration. If the preconditioner changes at every iteration, but was chosen offline (agnostic to the current iterate and gradient), then we can derive generalized sharpness using just the standard L2 sharpness, see line 206 and the subsection on Preconditioned $\ell_2$ Norm. But this isn't the case for $\ell_\infty$ and spectral norm, where the preconditioner depends on the current gradient, and is even a non-differentiable transformation of the gradient. Thus we can't really describe these methods non-Euclidean methods using a continuous coordinate transformation and the $\ell_2$ geometry
>
> ### Questions
>
> Q1: In the Euclidean setting, sharpness bounds local quadratic growth. In the non-Euclidean case, generalized sharpness provides a similar **non-smooth, non-quadratic** due to the used norm to measure distance. For our generalized sharpness
> $$S^{\|\cdot\|}(w)=\max_{\|d\|\le 1}d^\top \nabla^2L(w) d$$
> we have that the second-order Taylor expansion implies
> $$L(w+\delta) = L(w)+\langle \nabla L(w),\delta\rangle + \frac12 \delta^\top \nabla^2 L(w)\delta +o(\|\delta\|^2)$$
> $$\le L(w)+\langle \nabla L(w),\delta\rangle+\frac{S^{\|\cdot\|}(w)}{2}\|\delta\|^2+o(\|\delta\|^2)$$
>
> So $S^{\|\cdot\|}(w)$ measures the worst local second-order growth in norm $\|\cdot\|$. This non-smooth local model could be a good fit for the loss landscape when the loss is itself non-smooth. For instance, for $\|\cdot\|_1$, the $\ell_1$ sharpness $S^{\|\cdot\|_1}(w)$  captures how much the loss lanscape locally grows proportional to $\|\delta\|_1^2.$
>
> **Q2:** We refer the reviewer to our response to W1.

---

> > ### Author Rebuttal · Reviewer_EW6F · 2026-04-03
> >
> > I thank the authors for their responses. My concerns are resolved and I will consider raising the score.

---

### Official Review · Reviewer_zKDp · 2026-03-13

**Soundness:** 3
**Presentation:** 4
**Significance:** 3
**Originality:** 3
**Overall Recommendation:** 5
**Confidence:** 5

**Summary:**

The paper shows that non-euclidian GD methods also train on the corresponding edge of stability. In particular, they establish norm-dependent generalized sharpness, derive it for different norms and show empirically that it undergoes progressive sharpening, and then settles around $2/\eta$. They provide theoretical proves for convergence (if generalized sharpness <  $2/\eta$) and limited divergence, supporting this with a divergence for quadratic.

**Compliance With Llm Reviewing Policy:**

Affirmed.

**Final Justification:**

My concerns were mostly resolved - enough to raise the rating to a 5

**Key Questions For Authors:**

- Muon is not really used on its own in practice - with AdamW on embedding and output. How does it affect the training?
- I am a bit confused why directional smoothness is even introduced when there is the generalized sharpness, and why it is being measured. I can see it could be useful along the lines of discussion of Rayleigh quotient vs Hessian max eigenvalue, but this point should be further discussed
- For the experiments with high initial generalized normalized sharpness, can you start from a different initialization to make sure there is indeed progressive sharpening and then stabilization? If not, it should be discussed, too

**Limitations:**

yes

**Strengths And Weaknesses:**

# Strengths
- The paper is generally well-written: it gives a good overview of non-euclidian GD and the required prerequisites; a well-outlined step-by-step discussion for each norm of the construction of corresponding generalized sharpness and how it can be estimated; good coverage of norms/optimizers
-  Discussion of limitations of the divergence due to complexity of non-euclidian "maximizing directions". Due to this limitation, it is good to see a divergence being showcased on a quadratic approximation
- Interesting observation and discussion about pre-EoS oscillations
- Good comparison with the AEoS paper in the Appendix

# Weaknesses
- Do you have any experiments with cross-entropy loss? Whether yes or no, it should be explicitly mentioned
- Muon irregularitires
    - The range of learning rates of Muon is weird - it seems to be too small
    - Why the generalized sharpness in Muon so noisy? Is this an actual effect, or an artifact of Frank-Wolfe
- The whole “by definition” of progressive sharpening of directional smoothness — this is phrased very vaguely. It could easily be read that progressive sharpening happens by definition, but this is clearly not the case. Indeed we do not have an explanation behind progressive sharpening (e.g. Damian et al. "Self-Stabilization..."). Moreover, at very low step size/GF we never enter EoS phase, so that phrasing is not very precise.
- Equation (8) is incorrect I think you are missing the $(1-\tau)$ term there
- More minor, but I think there should be a discussion of what oscillations are, especially when mentioning there are oscillations that are not at EoS.

---

> ### Author Rebuttal · Authors · 2026-03-30
>
> We thank the reviewer for the positive assessment of our paper, highlighting the thorough treatment of non-Euclidean gradient methods and norms. We also value the recognition of our discussion of limitations, the illustrative use of quadratic approximations, the analysis of pre-EoS oscillations, and comparison with AEoS. We use W and Q to refer to Weaknesses and Questions
>
> ### Weaknesses:
>
> W1: All experiments with Transformer models use the cross-entropy loss.
>
> W2: We emphasize that the choice of LR is task-specific and varies with a model architecture and a data source. We also added LR for Transformer experiments, so all setups use three values: see **[[$\ell_{\infty}$](https://anonymous.4open.science/r/EoS_Figures-C8F6/L_inf_descent.png)]** and **[[Spectral](https://anonymous.4open.science/r/EoS_Figures-C8F6/SpectralGD.png)]**.
>
> For Muon experiments, we had to choose LR to both display EoS and still be convergent. In Fig 4, we used 8e-3, 16e-3, 32e-3 for CNN model, close to PyTorch default 1e-3 and choice in other works [1,2,3]. Using smaller LR resulted in the same noisy EoS dynamics or no EoS phase at all. Based on ablations in Sec F.2, G.2, H.2, we believe this noise isn't an artifact of FW, as FW estimation appears to be reliable (see also W1 to Reviewer EW6F for more info on this)
>
> [1] Liu et al, Muon is scalable for llm training, arXiv:2502.16982
>
> [2] Pethick et al, Training deep learning models with norm-constrained lmos, ICML 2025
>
> [3] Shah et al, Practical efficiency of muon for pretraining, arXiv:2505.02222
>
> W3: Yes, we agree that this "Thus, almost by definition, directional smoothness exhibits the sharpening and EoS phase" isn't precise enough. Instead, we wanted to say: "Thus, if the loss decreases monotonically and then oscillates, by definition directional smoothness must start below $2/\eta$ followed by EoS phase". As such, we do not explain why progressive sharpening happens, but draw a direct link between the dynamics of the loss and directional smoothness at EoS. We will make this more precise in the revision.
>
> W4: To make this argument cleaner, we now use mean-value form of the remainder for which we get $$ D^{\|\cdot\|}(w_{t},w_{t+1})=\frac{d_t^\top\nabla^2 L(w_t-\tau_t\eta d_t),d_t }{\|d_t\|^2}$$
> for some $\tau_t \in [0,1]$. The remaining arguments and text then follow verbatim.
>
> W5: Indeed, EoS oscillations should be clearly defined. We characterize EoS oscillations as persistent fluctuations in Hessian-related quantities (e.g., sharpness) around a specific threshold $2/\eta$, accompanied by step-to-step oscillations in the training loss, while the loss continues to decrease on average.
>
> As for the oscillations that are not at EoS, this is the subject of section B in the appendix and Fig 7. But thanks to the reviewer's question, we now realize we did not mention this section in the main text. We will correct this in the revision. We say that we observe oscillations that are not EoS when generalized sharpness is clearly below $2/\eta$ in Fig 7. We believe this type of oscillations is particular to non-Euclidean gradient methods, such as $\ell_{\infty}$ descent, where the sign of the gradient can flip from one iteration to the next, even before entering EoS.
>
> ### Questions:
>
> Q1: Indeed, the original Muon algorithm applies Adam updates to 1D parameters. In our experiments, we disable training of such parameters: we freeze them at initialization and optimize matrix-valued layers. This helps us to isolate EoS behavior of Muon, although it may lead to a suboptimal final performance, which is not the focus of our work. More generally, when different optimizers are used across layers, one can define an appropriate product norm (e.g., see [4] for examples) and then follow the framework we propose in this work.
>
> [4] Crawshaw et al, An exploration of non-euclidean gradient descent: Muon and its many variants, arXiv:2510.09827
>
> Q2: We used directional smoothness to arrive at our definition of generalized smoothness. That is, because we know directional smoothness has to start below $2/\eta$ and then oscillate around $2/\eta$, and because it is governed by the Hessian matrix, we suspected that the correct notion of generalized sharpness had to be closely related to directional smoothness (indeed, it's a tight upper bound). Without this bridge through directional smoothness, we were unable to guess what the correct definition of generalized sharpness should be. We wanted to share with the readers how we arrived at this definition of generalized sharpness. But the reviewer's point is valid, in that we need to make our motivation around directional smoothness more transparent. We will make this clearer in the revision.
>
> Q3: We provide additional experiments with Sign GD (normalized $\ell_{\infty}$-descent), where we initialize a model closer to 0, to clearly demonstrate the progressive sharpening phase **[[URL](https://anonymous.4open.science/r/EoS_Figures-C8F6/SignGD_initialization.png)]**.

---

> > ### Author Rebuttal · Reviewer_zKDp · 2026-04-06
> >
> > Thank you for the detailed response!
> >
> > I appreciate the clarifications - I think it addresses most of my concerns!
> > Q2 is good to know, Q1 clarification should definitely be added into the work (also that usually Adam is used for embedding/output layer - I assume you are using Muon there?), W3 is a good clarification.
> >
> > Two concerns remain: for Muon LR sizing, the CIFAR-10 speedrun uses LR around 0.24
> > (see https://github.com/KellerJordan/cifar10-airbench/blob/28bff5f5b31e95aa45b5b20e1f48baf1ed98d5f6/airbench94_muon.py#L362), which is quite different from the ranges you are testing on. Could you please include plots for much larger learning rate, even if there is no convergence (to showcase that when there is convergence, we do get stabilization of generalized sharpness at $2/\eta$). Also, could you comment on the spikes that are present in the SpectralGD runs?
> >
> > With that in mind, I am raising my score to a 5.

---

> > > ### Author Response · Authors · 2026-04-07
> > >
> > > We thank the reviewer for the follow-up questions.
> > >
> > > We note that CIFAR10-speedrun employs a specific architecture that differs from our standard 3-layer CNN with tanh activations. As such, one should not expect the same learning rate to perform optimally across both models. Nevertheless, we report results with LR=0.24. In this setting, the training loss does not decrease; however, the directional smoothness oscillates around $2/\eta$, while the generalized sharpness remains slightly above the threshold. We refer the reviewer to the requested figure: **[[URL](https://anonymous.4open.science/r/EoS_Figures-C8F6/Muon_LR_0.24.png)]**.
> > >
> > > We suspect that the spikes observed in Spectral GD are a variant of the “catapult” phenomenon [1]. Accordingly, our observations may be related to a non-Euclidean analogue of this effect. We leave a detailed investigation of these spikes to future work.
> > >
> > > [1] Lewkowycz, Aitor, et al. "The large learning rate phase of deep learning: the catapult mechanism." arXiv preprint arXiv:2003.02218 (2020).

---

### Decision · Program_Chairs · 2026-04-30

**Decision:**

Accept (spotlight)

**Comment:**

This paper studies the edge-of-stability phenomenon for non-Euclidean gradient methods and introduces a unified framework based on a generalized notion of sharpness. The work provides both theoretical insights and extensive empirical validation across a range of optimizers and model architectures. All reviewers found the paper to be technically sound, well-written, and relevant, highlighting in particular the clarity of the framework, the breadth of coverage across different geometries, and the practical contribution of estimating generalized sharpness via a Frank–Wolfe procedure.

Importantly, the authors engaged constructively with the reviewers during the rebuttal phase and were able to satisfactorily address essentially all concerns, including questions about experimental design, interpretation, and theoretical positioning. As a result, all reviewers either maintained or increased their scores, converging to a strong consensus in favor of acceptance. Overall, this paper makes a meaningful contribution toward understanding optimization dynamics beyond the Euclidean setting and provides a valuable perspective that is likely to stimulate further work in this area.